# Investigation of Smart Sustainable City Indicators of Sustainable Development—A Case Study of the City of Suwon

Robetmi Jumpakita Pinem [1,2] , Ancilla Katherina Kustedjo [3,*], Yelita Anggiane Iskandar [4]
and Bernardo Nugroho Yahya [5,*]

1. Department of Business Administration, Universitas Diponegoro, Semarang 50275, Indonesia;
robetmi@lecturer.undip.ac.id
2. Department of International Trade, Kangwon National University, Chuncheon 24341, Republic of Korea
3. Association of Indonesian Researchers in South Korea, Seoul 07342, Republic of Korea
4. Department of Logistics Engineering, Universitas Pertamina, Jakarta 12220, Indonesia;
yelita.ai@universitaspertamina.ac.id
5. Department of Industrial & Management Engineering, Hankuk University of Foreign Studies,
Yongin 17035, Republic of Korea
* Correspondence: ancilla.kustedjo@gmail.com (A.K.K.); bernardo@hufs.ac.kr (B.N.Y.)

**Abstract:** This paper aims to investigate the interlink between city sustainability development indicators and smart sustainable city indicators (SSCIs). While the sustainability development indicators of a city mostly rely on its sustainable development goals (SDGs), no investigations into whether these indicators imply SSCIs have been conducted. SSCIs are critical during the current unprecedented climate crisis and are necessary to facilitate a livable future. This study considers the city of Suwon as a case study. Suwon's sustainable development policy adopts 10 SDGs among the 17 SDGs from the United Nations, later called Suwon SDGs (SSDGs). By conducting a content analysis, this study found that the indicators adopted from the SDGs require further investigation to evaluate the core SSCIs. Using text analytics, we found that Suwon's policies focus on indicators in the local environment, such as the employment rates for every age category, even if the SSDG indicators incorporate the SDGs. The purpose of this analysis is to gain more attention from policymakers about the necessity of reconstructing and considering SSCIs as a part of a smart city's sustainable development. Lessons and practical implications are discussed for future studies.

**Keywords:** smart city; smart sustainable city indicators; sustainable development; sustainable development goals

## 1. Introduction

The development of ICT has an impact on every aspect of the world. It affects not only general aspects such as human life but also business and management aspects such as logistics. Logistics services are one of the most important domains in daily life. They are also important for business organizations since a chain is required to transfer materials, goods, and equipment from suppliers to consumers. While speed and cost are the key factors of logistics success, the environmental and social aspects of logistics are commonly negligible. ICT system integration is necessary to achieve economic efficiency along with sustainable logistics by combining government, environmental, social, and economic goals through smart sustainable logistics [1].

Governments have proposed the idea of "smart cities" as a way to address societal issues and improve people's quality of life in general [2]. They are classified as technologically sophisticated metropolitan areas that rely on innovation to promote competitiveness, sustainability, and a high standard of living [2]. Numerous aspects are relevant in the context of sustainable logistics since this industry offers a unique chance to promote sustainable practices. The implementation of environmental and social programs by suppliers,

the decrease in plastic or box packaging, the improvement of warehouses' working conditions, the adoption of fuel-efficient transportation modes, and other initiatives are notable examples of these types of actions [3].

Korea has become a key actor in developing smart cities, with the South Korean government actively pursuing smart city initiatives throughout numerous locations. These initiatives include a number of cutting-edge high-tech solutions, such as building information management (BIM), smart traffic monitoring systems, and artificial intelligence of things (AIoT). The adoption of these initiatives shows potential for tackling a range of urban difficulties, including those related to mobility, the environment, and energy efficiency. Recognizing the paradox that results from relying entirely on high-tech solutions is crucial since these solutions might not always be consistent with the environmentally responsible actions that are crucial for sustainability. Other aspects, such as familiarity with commuter habits and ecological consciousness, may provide workable solutions for developing sustainable logistics in these situations. For instance, the use of renewable feedstocks such as polylactic acid (PLA) in the production of biodegradable plastics and the design of public transportation routes based on social dynamics can contribute to sustainable practices.

With a focus on sustainability, the United for Smart Sustainable Cities (UNECE) program seeks to advance our knowledge of smartness in urban planning. The term "people-smart sustainable cities" refers to the dedication of cities to fill capacity and efficiency gaps, address social needs, and promote innovation. Previous work has discussed the creation of indicators for smart sustainable cities as an extension of the SDGs. While many cities attempt to utilize the SDG indicators, few of them consider smart sustainability as a development factor.

This study aims to investigate how city sustainable development indicators leverage smart sustainable city indicators. Existing smart sustainable cities are focused on the conceptual framework [4,5], and there is a lack of investigation into how the proposed indicators and the conceptual framework are applicable to the indicators of sustainable development cities. This study emphasizes the use of a case study and focuses on Suwon, as one of South Korea's smart city projects. In terms of its triple bottom lines, economy, society, and environment, this study attempts to answer questions about whether the proposed city sustainable development goal indicators align with smart sustainable city indicators. By looking at the similarities of each indicator both syntactically and semantically, this study contributes the following:

-　Provides an approach to evaluate city SDG indicators with the SSCIs.
-　Evaluates the alignment between the Suwon SDG (SSDG) indicators and the SSCIs.
-　Enhances the SSCIs and suggests that main stakeholders adapt the core indicators for the development of smart cities.

The results of this study enrich the literature and could be used as a reference for various projects related to smart sustainable city programs.

## 2. Literature Review

### 2.1. Smart City

The concept of a "smart city" can offer developments and solutions that make it easier for the general population to obtain information quickly and effectively [6]. The city is also capable of effectively and efficiently managing all resources to assist and resolve issues with comprehensive, ground-breaking, and long-lasting solutions for high-quality city services [7,8]. The term "smart city" has two definitions according to the literature review: (1) a city that incorporates ICT technology [9,10]; and (2) a new method of city planning in which the natural world, education, and human and social capital all play significant roles [11–13]. As a result, we have come across definitions that emphasize the technological aspects of smart conceptions. Smart cities according to Bakici et al. [14] use new technologies to create competitive, advanced, tech-intensive, sustainable green cities with innovative commerce and an increased quality of life. Rodríguez-Bolívar [15] explained that smart

cities can be conceptualized in different ways, from bringing together complex information technology (IT) in urban contexts to creating organizational capabilities, constructing physical infrastructure, and developing innovative services.

IT has never been more important for leveraging creativity, new technology, and city resources to turn obstacles into opportunities for smart cities. According to Ojo et al. [16], smart cities should be researched and examined in terms of a few different factors [17–20]: (1) institutional rules and other arrangements; (2) governmental services; (3) management and administration; (4) a knowledge-based economy and business-friendly environment; (5) governance, engagement, and collaboration; (6) developed environment and city facilities; (7) human capital and creativity; (8) ICT and other technologies; (9) natural surroundings and ecological sustainability; and (10) data and information. Aspects that have been impacted by supply chains/logistics in other domains are suggested in other works in the literature. Tachizawa et al. [21] conducted an empirical analysis of the supply chain affecting smart city operations. Diverging from the current focus on education, infrastructure, healthcare, and energy, Lee et al. [22] showed via a sector-based analysis that smart transportation can have an effect on smart city operations. According to Dameri and Ricciardi [23], smart cities need sustainability, resilience, and a high quality of life, and the key resources required for smart cities' intellectual capital are listed in Figure 1. This consists of environmental capital and institutional capital, units of analysis, territorial systems (i.e., transportation or waste), and the key managerial challenges that result from these.

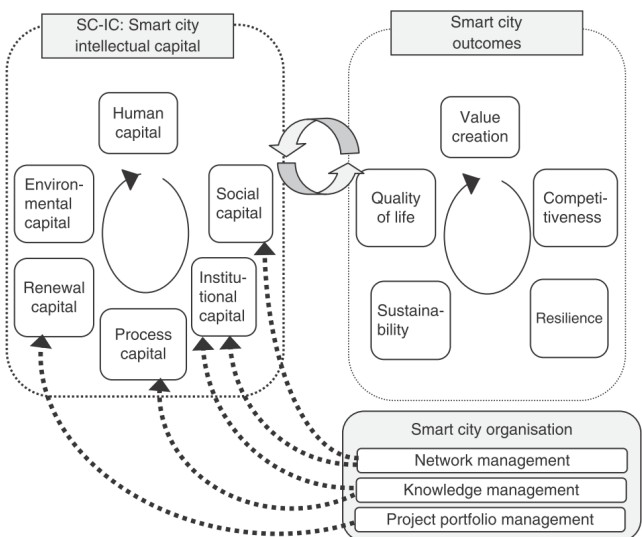

**Figure 1.** Framework SC-IC (Smart City-Intellectual Capital) [24].

## 2.2. Sustainable Development

Sustainability is a thinking paradigm in the process of urban development [5]. When forming policies, sustainability is considered alongside the three bottom lines of economic development, social equity, and environmental protection [5]. The Organization for Economic Co-operation and Development (OECD) described some additional sets of sustainable development elements such as prosperity/profit, people, the planet, peace, and partnerships [24]. However, a lot of studies still utilize the common conceptual framework of social, environmental, and economic dimensions [25].

Sustainability indicators have been investigated by many researchers and organizations. The authors of [26] developed a set of policies and procedures for sustainable development projects. Their focus was only on the steps to achieve the goals of sustainability without the provision of indicators. The three dimensions of sustainability—social (access, equity, health, and safety), environmental (climate change, air quality, noise, land use, biodiversity, waste), and economic (development, efficiency, employment, competition,

and opportunity)—have been discussed in detail [3,27,28]. Other authors have published a working paper for the OECD targeting the sustainable development indicators used by national and international agencies. Their results showed a set of indicators categorized based on four aspects: social, environmental, economic, and institutional pillars, which included 15 themes and 38 indicators [24]. These indicators were the basis for the smart sustainable city indicators developed by [4].

### 2.3. Smart City and Sustainable Development

Recently, many works have attempted to investigate the relationship between smart cities and sustainability. In the context of practical implementation, some challenges for smart sustainable cities have been noted, such their excessive amount of attention on technology, practice complexity, and ad hoc conceptualization [29]. A paper on developing the taxonomy of smart sustainable cities was noted as the first to explore the non-technological aspects of smart cities [4]. It also offered a set of mutual indicators between sustainability and the smart city concept instead of focusing on the implementation of smart city initiatives in urban areas [4].

Figure 2 displays the framework for balancing the three dimensions of sustainability in a data-driven, smart sustainable city. Environmental sustainability is illustrated as the capability of the environment to maintain its function continuously. The purpose of environmental management is to prevent and stop actions that lead to environmental deterioration, as well as to reduce its effects. Economic sustainability is characterized as an approach to achieving economic growth while preserving the environment. This approach involves finding a way to minimize natural damage and building up the earth's resources in a sustainable way. Social sustainability is described as a compilation of performance and achievements to encourage development. Instead of reducing the availability of human resources, this helps to increase their capacity. Furthermore, social sustainability involves the creation of harmonious and sustainable communities.

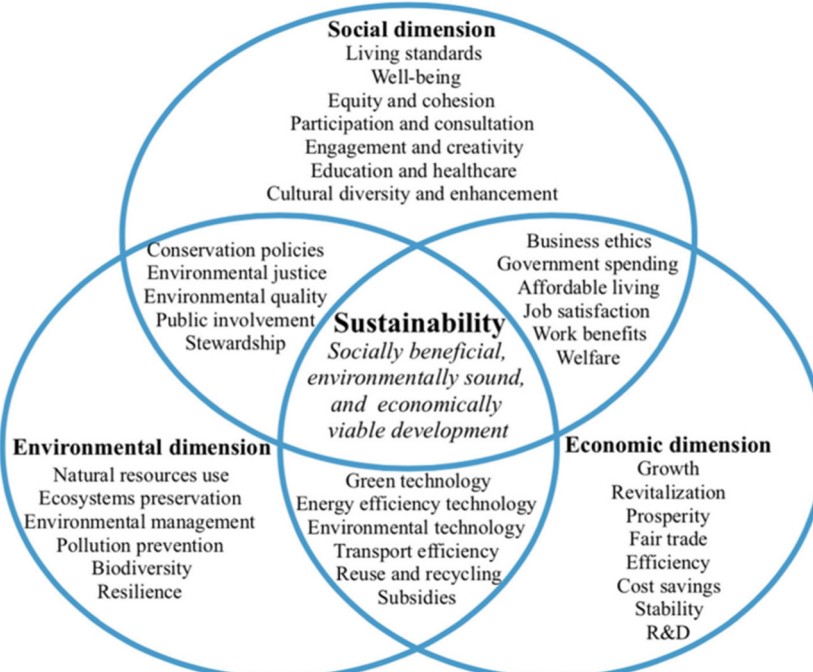

**Figure 2.** A data-driven smart sustainable city for balancing the three pillars of sustainability [5].

Looking at the commonality of smart cities and sustainability, there are several dimensions of sustainability that cover the dimensions of a smart city, such as citizen engagement, the need for responsible resource management, and energy efficiency [30]. Researchers have attempted to preserve smart cities' dimensions to address sustainable development.

Other works [1,2] have thoroughly assessed the literature on smart sustainable cities and discussed the benefits and challenges of developing such cities. One study identified 19 research gaps in the study of smart sustainable cities. It stated that the gaps show a need for a collection of standard indicators that encompass both smart cities and sustainability concepts. While smart sustainable city indicators have been proposed, these concepts should be investigated properly to prove their practicality and applicability in terms of sustainability and smartness [31]. The current work aims to explore the commonality between the indicators of smart sustainable cities and sustainable development cities. The indicators of smart sustainable cities are taken from [4] whilst the applicability of the indicators is taken into account via one case study of Suwon, one of the dedicated smart cities in Korea [32].

### 2.4. Bibliometric Analysis of Smart Cities and Sustainability

The bibliometric synopsis is presented via an intricate visualization in this paper. An analysis of research development trends in smart cities and sustainability was carried out for publications up to 2023. The three main keywords, "smart city", "sustainability", and "case study", were combined when finding text-data-based publications. We collected data by performing searches via the Crossref, Google Scholar, OpenAlex, Pub Med, and Semantic Scholar databases. These word queries were investigated within the article title, abstract, and stated keywords by the authors. We found thousands of papers that matched all our determined queries, and had been published within the last 10 years (since 2013). If we look at the network visualization in Figure 3, specifically for the term "Korea", we can only see connections to competitiveness, republic, association, and one area in the country, Songdo. There is no study yet that connects smart city subjects to sustainability for the city of Suwon. If we take a further look at the terms similar to "case study", such as "case study analysis", "multiple case study", "case study area", and "real case study", we cannot find any studies that combine each of these terms for a specific, observed city (Figure 4). There has likewise been no discussion of the city of Suwon and its smart city policies. There was only one term, "study area", that connected China and land (Figure 5). We looked deeper by using the name of a city, country, or even continent to investigate their connection to sustainable smart city analyses. Via this bibliometric analysis using the VosViewer software version 1.6.19, we found that there have not been any previous studies with our same idea. In Figure 6, we mapped the publications' distribution years and found that the most relevant papers were published quite recently, from 2017 to 2019.

Suwon is the capital and largest city of Gyeonggi-do, South Korea's most populous province surrounding Seoul, the national capital. The total area is 121 km$^2$ with a population of 1.2 million people or 9.3% of the Gyeonggi population in 2022 [33]. Suwon has a master plan to better develop the city into a hub of education, research, and industry in the southern part of the capital region [34]. As part of the plan, 14 housing sites have been developed, as well as the Suwon Industrial Complex (SIC), which specializes in the IT, electronics, and mechanical industries. Neighboring cities substantially depend on Suwon's banking and insurance industries, along with public infrastructures and services. As such, discussions have been held regarding the integration of Hwaseong and Osan into Suwon's administrative boundaries [35]. The population growth rate in Suwon has been 4.8% annually since Samsung Electronics' establishment in the late 1960s, and the gross regional domestic product growth rate has been 18% since the 1970s [36]. An R&D center, Samsung Digital City (SCD), was built by the company in the 2000s, replacing the blue-collar workers in the city with highly educated workers. As a result, more residents outside of Suwon have moved there, and the city has become younger, with the majority of the population in their 30s and 40s [36]. In addition, the city has developed housing planning sites in the Gwanggyo New Town (GNT) and Homaesil residence zones as a part of its plans to become a smart city [33].

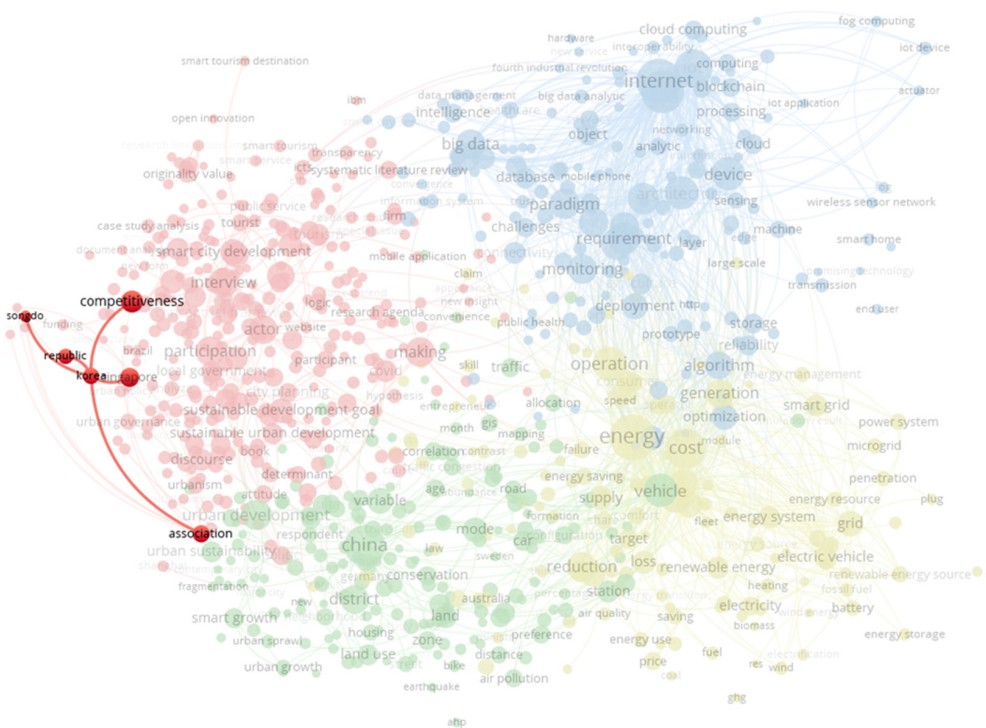

**Figure 3.** Complete Network Visualization.

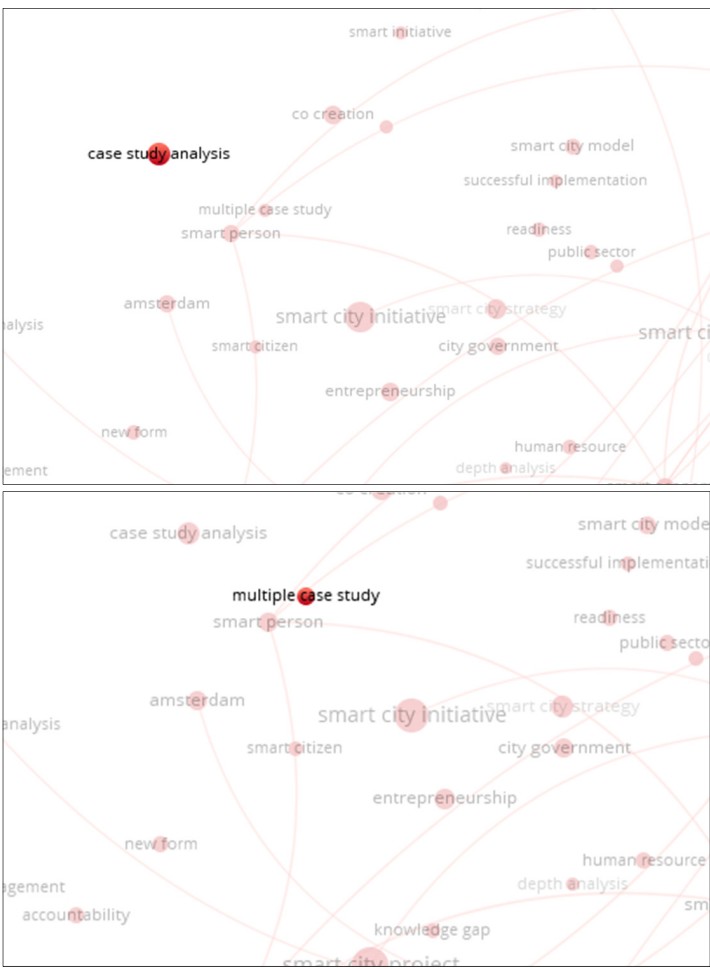

**Figure 4.** Network Visualization: Case Study or Similar.

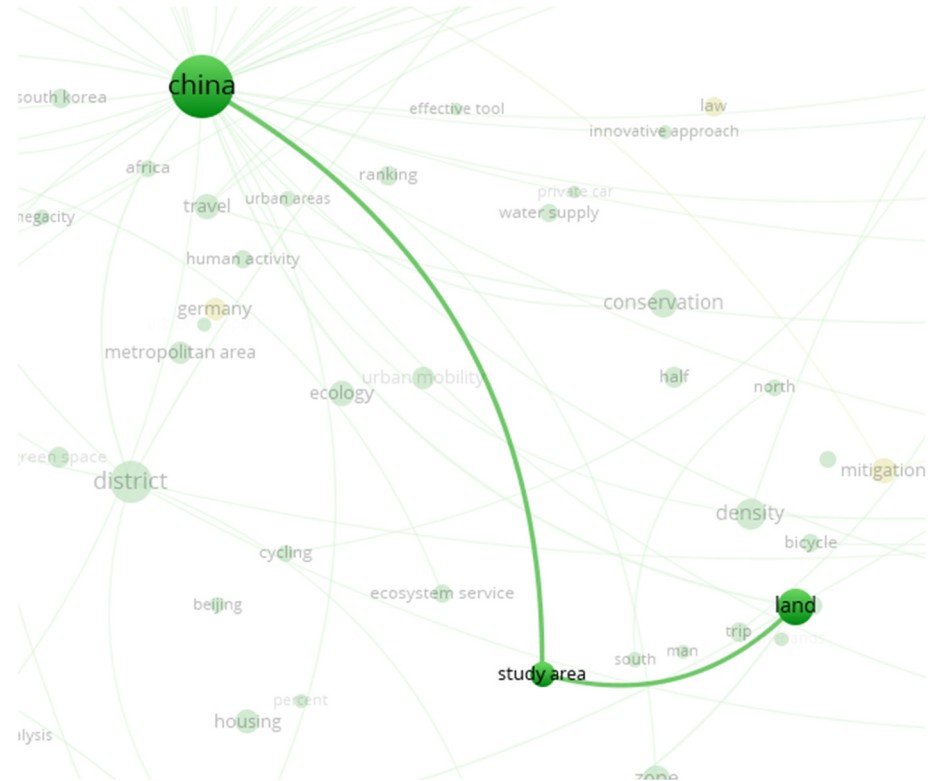

**Figure 5.** Network Visualization: China Case Study.

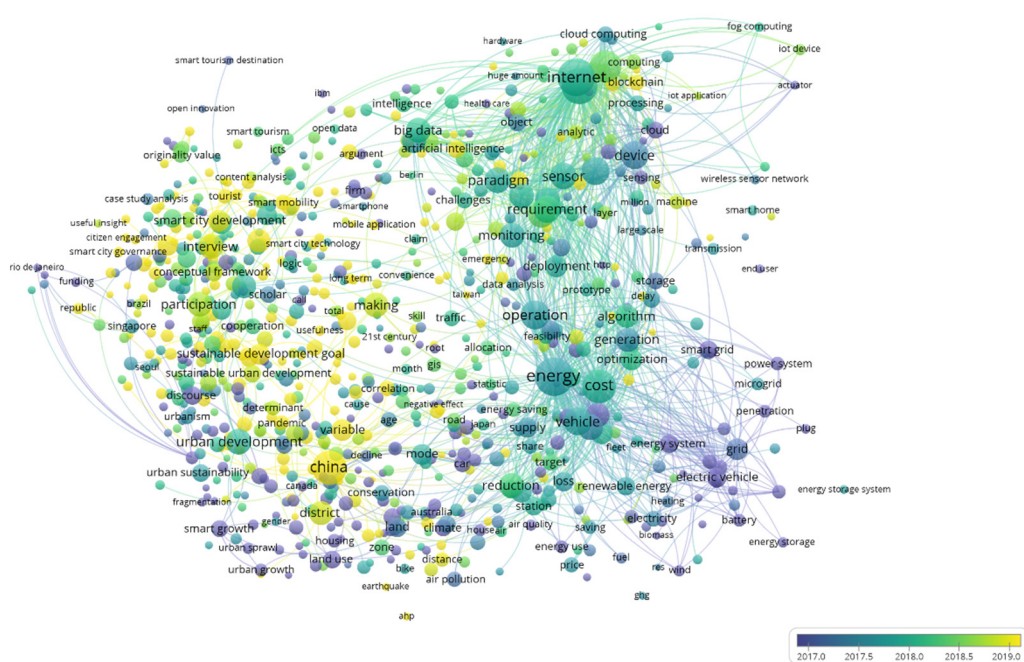

**Figure 6.** Overlay Visualization Map2.5. The City of Suwon.

In the first phase, Suwon developed a five-year plan (2011–2015) that focused on managing land use, traffic, the environment, administration, finance, and information communication comprehensively. As part of the second phase, the city of Suwon's smart cities key initiative, the Korean Ministry of Information and Communication in partnership with the Ministry of Construction and Transportation established a working group to handle issues related to ubiquitous cities (U-cities) in newly formed communities. In 2018, the city

of Suwon participated in the 'City Selective Demonstration Project for Smart City National Strategy Project; Promoted by the Ministry of Land, Infrastructure, and Transport and Ministry of Science and ICT' [32]. In addition to common core technology development tasks, the Smart City National Strategic Project contains two factual research projects: one aimed at solving urban problems such as traffic, safety, and city administration; and the other aimed at creating businesses by applying innovations in industries related to energy, the environment, and welfare. In 2022, the city of Suwon established an urban transportation maintenance plan [37], a core strategy for local traffic safety for improving pedestrian safety and personal transportation convenience. There are three primary objectives for the city of Suwon: developing a safe environment for residents in areas with heavy traffic, measuring traffic safety in the transportation industry, and increasing public awareness of traffic safety. Further, the city of Suwon's vision is of a "smart transportation city where people are happy", and its maintenance schedule for urban transit is designed to achieve three goals: smart transportation that anyone can use anytime, safe and comfortable transportation that is centered on people, and convenient, eco-friendly public transportation that is preferable to private vehicles.

## 3. Research Methods

This study uses two kinds of methodology: case analysis and content analysis. Case analysis, that is, a case study, aims to gain practical and context-dependent knowledge to understand a real-life context [38]. In addition, a case study is beneficial for exploring a set of contemporary practices over which researchers have little or no control and can empower researchers to argue for or against theory more rigorously [38].

This article aims to examine and assess how Suwon as an urban city implements the Sustainable Development Goals (SDG), later called Suwon SDGs (SSDGs) [39] by taking into account 10 points—namely, whether it has environmentally friendly energy available for every climate; healthy and harmonious biodiversity; a clean and clear water cycle; a healthy, sustainable food supply; industrial innovation and excellent employment; sustainable consumption and production targets; welfare, health, and education for the happiness of its citizens; gender, equality, diversity, and culture; and autonomy, justice, peace, and harmony through self-governance.

Content analysis [40] aims to analyze the two cohorts of indicators. This technique is frequently used to determine the similarities between two sets of variables that have been gathered through interviews, surveys, or the use of secondary data [41]. In this study, we employ content analysis methods in two phases: quantitative and qualitative. The quantitative method analyzed the lexical relationship among the words by evaluating the term frequency in the sets of indicators and calculated the cosine similarity based on the vectorization of the sentences using term frequency–inverse document frequency (TF-IDF). The high value of TF-IDF is utilized for the qualitative method to find the semantic correlation between words.

In detail, the content analysis contains five steps as follows:

- Content selection: It uses two sets of indicators (the SSDGs and the SSCIs).
- Text Preprocessing: It uses the common procedure (i.e., tokenization, the removal of stop words, stemming) to remove the commoner morphology from words in English.
- Filter: It sets a threshold value to filter the most similar indicators based on the TF-IDF using the cosine similarity of the text analysis.
- Coding: The researcher manually analyzes the text.
- Analysis: It discovers the indicators based on the correct semantic and lexical relationship of words.

The SSCIs proposed by [4] offered a combination of sustainable development indicators suggested by the OECD and the Smart City Index Master by Cohen [42]. The resulting set of indicators and 28 associated indicators are used in our study as the reference. Since there are many common words in the indicators, we employ text analytical approaches to evaluate the similarity of the words. Prior to the text analysis, we perform text pre-

processing by tokenizing the indicators, removing the stop words, and stemming. The preprocessed data is converted into TF-IDF feature and the similarity technique, that is, cosine similarity, is used to find similar indicators. Existing software is available to complete the coding step [43,44]. Instead of using software, in this study, we review the filtered contents manually since the data become smaller after the similarity measurement based on the text preprocessing. Finally, an analysis is conducted to seek the conceptual correlations between the SSDGs [39] and the SSCIs [4]. The reason for the final analysis is to investigate the indicators in SSCIs that imply SSDGs and vice versa.

## 4. Results

This section aims to provide an overview of the city of Suwon. Afterward, the Suwon Sustainable Development Goals based on the indicators developed by SDGs will be described in detail. Subsequently, the content analysis is utilized to investigate the correlations between the SSDGs and SSCIs. In the end, we discuss the results of the content analysis.

In this study, the city of Suwon claims that encouraging world peace and good health can lead to sustainability (i.e., in terms of economic, social, and environmental performance). Suwon is moving towards sustainability by developing the Suwon Sustainable Development Goals (SSDGs), identifying the appropriate indicators to evaluate its progress, and ensuring their effectiveness in implementation and monitoring. The city of Suwon is dedicated to achieving sustainable development as part of its vision to create a people-friendly city. Global cooperation for sustainable development should be based on sustainability diversity, respecting the diversity of cities rather than seeing them uniformly and setting sustainable development goals appropriate to each city's needs.

Several studies have been conducted in the city of Suwon. Jang and Yoon [45] examined the perceptions and attitudes of residents of Gwanggyo New Town toward the Suwon Convention Center's development. Their study showed that residents' perceptions of the convention center's development were dominated by its social, cultural, and economic impacts. Furthermore, their results indicated that socio-cultural and economic factors had a significant impact on the residents' attitudes towards development, while environmental factors had the lowest impact. They excluded variables such as the occupation, economic power, knowledge, and income level of the residents, which made it difficult to verify the social exchange theory. Furthermore, some second-generation new towns have embraced smart city solutions, to project the image of pursuing environmental sustainability and high-quality living environments, in addition to developing high-tech clusters such as techno valleys (e.g., Gwanggyo Techno Valley) [46]. Moreover, Ryu and Lim [47] described the city of Suwon as a smart city, utilizing smart city services to manage and detect old infrastructure. It also plans to utilize sensing technologies, 3D urban maps, and a geographic information system (GIS) to detect an invisible underground water supply. Moreover, the city of Suwon established an online "SOS-net-system" for local businesses to request assistance to maintain economic sustainability. The development of its technological infrastructure was driven by government-led projects during the early 2000s, which focused on establishing hardware infrastructure.

### 4.1. Suwon Sustainable Development Goals

Suwon places residents' well-being at the center of its aim to build a people-friendly community in efforts to improve life quality. Suwon residents are given the opportunity to take part in policymaking and collaborate with other people and the local administration. As a consequence, Suwon's sustainability strategy can be summed up in terms of three characteristics: environmental, social, and economic performance. Suwon believes that city sustainability cannot be guaranteed without the direct participation of citizens. In this case, the city has made a commitment to fostering "participatory governance", in which locals actively engage in urban planning, identify problems, carry out projects, track advancements, and assess overall policies. Consequently, a vision for the "Sustainable City of Suwon" was created by the Suwon Council for Sustainable Development. Moreover, the

council has drawn up habitat conservation plans. A sustainable city is better understood by various stakeholders, such as schools, community organizations, businesses, and public entities. The city of Suwon has adopted 10 SDGs comprising three main aspects: the environment (clean energy for all climates; healthy and harmonious biodiversity; and a clear and clean water cycle), economy (healthy and sustainable agriculture and foods; excellent employment and innovative industries; and sustainable consumption and production), and social (welfare, health, and education for the happiness of its citizens; gender, equality, cultural diversity, resilience, and involvement in culture; and independence, justice, peace, and coherence through self-governance). The details of the 10 SDGs have been discussed in the report [32] and are summarized in a later section.

### 4.1.1. Clean Energy for All Climates

**SDGs of Affordable and Clean Energy.** Suwon has been implementing the following policies for sustainable environmental capital to create a carbon-neutral city since 2010. To increase the energy efficiency of public institutions, the city has been developing initiatives and regulations to support renewable energy in the public sector. In the private sector, the city has been working on projects including building a shared solar power plant, installing photovoltaic power using facilities for resource recovery, producing small amounts of solar power, and building solar-powered homes. In addition, the city has been collaborating with groups from civil society, such as the Climate Action Network, to make its transportation system more eco-friendly. To improve air quality, it supported the introduction of eco-friendly vehicles and electric buses and encouraged bicycle commuters to foster a dust-free community. Furthermore, the city is the only local government in Korea that oversees greenhouse gas (GHG) inventories and establishes yearly targets for carbon emissions per capita and energy consumption that encourage energy efficiency, eco-friendly buildings and transportation, and renewable energy projects. Moreover, the city has been working to improve its capacity for climate change adaptation by reducing its average temperature by one degree Celsius through heat island mapping and the management of heat-prone areas. Through the city's collaboration with organizations from civil society, it plans to plant 100 million trees, increase its green space, and decrease traffic loads. Suwon established the following seven targets in Table 1 as part of its response to climate change.

**Table 1.** Clean energy for all climate adaption targets.

| No. | Target | Description | Indicator |
|---|---|---|---|
| A1 | Energy security and efficient energy generation | 25% energy independence should be attained. | 1. City's electricity independence rate (%).<br>2. Renewable energy in the city's total annual power production (%).<br>3. The number of citizens who participate in power generation. |
| A2 | Smarter urban planning and energy conservation | Reduce energy consumption through conserving energy and fostering efficiency. | 1. City's power consumption per capita.<br>2. The number of green village households. |
| A3 | Energy welfare | Community benefits via distributing electricity. | The yearly accumulated value of energy "Sunlight Sharing Solar Power Plant". |
| A4 | Promoting eco-mobility while enhancing air quality | Reorganize the city's transportation system to be more eco-friendly and improve air quality. | 1. The quantity of green electric vehicles registered.<br>2. Share of public transportation (%).<br>3. The number of days with a particulate matter (PM) measurement over 2.5 in a year ($\mu m/m^3$ (24 h)). |

**Table 1.** *Cont.*

| No. | Target | Description | Indicator |
|---|---|---|---|
| A5 | Reducing carbon emissions | Lower the city's overall energy consumption and carbon emissions per capita. | Greenhouse gas emissions of Suwon. |
| A6 | Reducing city temperatures and enhancing the city's ability to respond to climate change | Reduce the average temperature by 1 °C, map the city's heat island intensity levels, and manage heat-sensitive zones. | Days with extreme weather conditions (i.e., heat waves, tropical nights, average annual temperature, etc.). |
| A7 | Smarter urban planning and energy conservation | Control demand, reduce use, increase efficiency, and save energy. | 1. The amount of Green Climate Fund and the number of communities that participate in climate actions. 2. The number of participants in educational programs on climate change. 3. The number of climate change programs participated in by citizens. 4. The implementation rate of action plans of the city's climate change adaptation measures. |

4.1.2. Healthy and Harmonious Biodiversity

**SDGs of Life below Water and Life on Land.** Due to the surrounding natural environment of the mountain Gwanggyo in the north and the mountain Chilbo in the west, the city of Suwon adopted the motto "Healthy and Harmonious Biodiversity". Residents of Suwon take part in institutional, physical, and technical initiatives to live in harmony with nature. To stop haphazard growth, save green spaces, and encourage eco-friendly urban life, city administrators and actors from civil society have collaborated on public awareness campaigns and educational programs. Suwon established the following five goals, listed in Table 2.

**Table 2.** Healthy and harmonious biodiversity targets.

| No. | Target | Description | Indicator |
|---|---|---|---|
| B1 | In the city, eight key flagship and indicator species are being monitored | Observe and share information about eight major flagship species in all villages and indicator species in climate-change-affected areas, as well as compile reports on shifting trends. | Eight key city flagship species and available monitoring stations for indicator species in climate-change-affected areas. |
| B2 | Protecting and expanding landscape conservation areas to protect their ecosystems | Increase the number of wetlands and ecological conservation areas and study habitats for species, as well as produce reports on climate change. | 1. The total size and the number of landscape conservation and wetland conservation areas. 2. The number of areas considering species recovery plans. |
| B3 | Expansion of natural areas | Increase the proportion of natural areas and urban forests that meet the national recommended levels, focusing on first-grade areas. | 1. Natural area rate (%). 2. Area of urban forest per capita (%). |

| No. | Target | Description | Indicator |
|-----|--------|-------------|-----------|
| B4 | Education on biodiversity and promotion of citizens' awareness | Enhance citizens' understanding of biodiversity in city's historical and unique areas, take care of children's future ecological efforts, build a better environmental education system, and create a more eco-friendly city. | 1. The number of schools with educational programs on biodiversity. <br> 2. The number of facilities and organizations supported by the government promoting biodiversity awareness. |
| B5 | Civic involvement and governance for biological habitat conservation | Continue to establish citizen engagement programs for biological habitat conservation and to foster private–public governance for biodiversity. | 1. The number of citizens involved in conservation activities per year. <br> 2. The number of biodiversity-related policy initiatives implemented each year by private–public governance. |

### 4.1.3. Clear and Clean Water Cycle

**SDGs of Clean Water and Sanitation.** Suwon, the city's name, literally means "water sources", and the city plays an important role in delivering water to nearby counties/cities. The Hwanguji Stream transports the waters of its principal streams, which run into streams in neighboring cities, such as Jinwi, Anseong, and Pyeongtaek. Water management has always been valued by the city since how the city controls its water quality and quantity affects surrounding communities as well. Suwon established a wide variety of policy objectives and tried techniques for controlling the ecological status of its four major streams, non-point source water pollution self-sufficiency rates, and the management of impermeable layers and groundwater in 2020. Then, Suwon set the following seven targets to become a city with a clear and clean water cycle, as shown in Table 3.

**Table 3.** Clear and clean water cycle targets.

| No. | Target | Description | Indicator |
|-----|--------|-------------|-----------|
| C1 | Maintaining healthy river habitats and monitoring aquatic life | Preserve longitudinal–horizontal linkage to sustain healthy river ecosystems, continue to investigate river ecosystem conditions, and expand the number of monitoring points. | 1. Stream naturality grade from major streams. <br> 2. The amount of the city's budget allocated for a biological survey of streams. <br> 3. Rate of growth in biodiversity in main streams in terms of species. <br> 4. The number of official monitoring locations (hot spots) for assessing the environment of streams. |
| C2 | Enhancing the water quality of streams and lakes | Lessen non-point source pollution, reduce pollution load to meet national criteria, and carry out non-point source pollution-related tasks. | 1. The rate of reduction in pollutant load. <br> 2. The water quality of four vital rivers and lakes. |
| C3 | Increasing public knowledge in order to establish a water management system that includes community involvement | Expand education on the ecological perspectives of the water environment and urban rivers in order to create a water management system in which citizens can participate. | 1. The number of educational programs on the environment and circulation of water. <br> 2. The amount of water environment and circulation education programs. |

**Table 3.** *Cont.*

| No. | Target | Description | Indicator | |
|---|---|---|---|---|
| C4 | Increasing water self-sufficiency and conserving water | Increase water self-sufficiency and implement policies regarding rainwater use, water recycling, and water conservation. | 1.<br>2. | Water self-sufficiency rate.<br>Per capita water consumption. |
| C5 | Expanding regions with permeable surfaces to allow for rainwater infiltration | Reduce areas with impermeable surfaces to optimize water circulation. | 1.<br>2. | The percentage of the total area covered by permeable surfaces.<br>Annual average groundwater level. |

### 4.1.4. Healthy and Sustainable Agriculture and Foods

**SDGs of Zero Hunger (No Hunger).** As a densely populated city with an ongoing appetite for food and development, Suwon wants to encourage urban agriculture and supply healthy food. The city has allocated almost KRW 100 billion for free meals for children from four and five years old to high school students. The Basic Food Plan of Suwon suggests combining school meals with public meal services in the social welfare sector. 'The Integrated Food Strategy' necessitates the entire food manufacturing, transportation, consumption, and disposal process. It integrates sustainable development concepts and values such as environmental protection, safety, and health. Due to its urbanization and high food consumption, the number of farming households and agricultural areas has decreased gradually in the city of Suwon. In addition, production costs are high, so land prices have steadily grown in conjunction with the rising demand for residential development. The city also prepares budgets to improve food quality and expand school meals, as well as support urban agriculture and a healthier dietary lifestyle. Moreover, Suwon has the largest urban farming system in Gyeonggi Province, and its residents are engaged in urban farming. The city also promotes local food and its accessibility and increases farmers' income by offering direct food markets in traditional markets, such as the market around Mount Gwanggyo. The following four targets have been set by Suwon as part of its efforts for healthy and sustainable agriculture and food, as described in Table 4.

**Table 4.** Clear and clean water cycle target.

| No. | Target | | Description | Indicator |
|---|---|---|---|---|
| D1 | Creating a local food system and its management | 1.<br><br>2.<br><br><br><br><br>3.<br><br><br>4. | Create a long-term local food system based on equity, safety, health, and care.<br>Encourage eco-friendly and local food values throughout the food production, distribution, consumption, and recycling processes.<br>Develop and implement the "Basic Plan on Food for Citizens" as a food and agricultural policy.<br>Submit public annual progress reports on the implementation of the basic plan in the form of a food policy committee/council. | The rate at which the Basic Plan on Food for Citizens is being implemented. |

**Table 4.** *Cont.*

| No. | Target | Description | Indicator |
|---|---|---|---|
| D2 | Ensure citizens' right to food | 1. Make sure that all residents have access to nutritious and sufficient food.<br>2. Provide low-income residents with public food service.<br>3. Provide food to the young out-of-school population.<br>4. Ensure that no citizen suffers from social hardship or hunger. | The number of citizens benefiting from public food service. |
| D3 | Improve local farmers' and food producers' income | 1. Increase the income of local farmers and producers of processed food.<br>2. Increase the number of direct markets and retailers for locally produced agricultural products.<br>3. Encourage the adoption of smart labels by local farmers and food goods.<br>4. Support local food processing and distribution to help local SMEs.<br>5. Find new markets for locally produced agricultural products. | 1. The average income of farming households.<br>2. The number of direct local food markets and businesses, as well as their annual turnover. |
| D4 | Promote citizens' knowledge of food and healthy eating culture | 1. Sustainable local healthy food systems.<br>2. Educate citizens on healthy eating.<br>3. A well-functioning food inspection system ensures safe food that is free of radiation, pesticides, and GMOs. | 1. The number of people who have participated in food education.<br>2. Amount of the city's budget for food safety inspections. |

### 4.1.5. Quality Jobs and Industrial Innovation

**SDGs of Industry, Innovation, and Infrastructure.** The city of Suwon ensures the sustainable development of citizens' livelihoods by promoting stable employment and safe working conditions. The following metrics are used to gauge its progress: the creation of high-quality jobs; improvements in wages and working conditions; friendliness to start-ups; the establishment of a proper social-economic ecosystem; and the encouragement of SMEs. From 2015 to 2019, the number of people who had jobs in Suwon continuously climbed to 521,000. Currently, the employment rate stands at 60%, and the employment rate for young people (15–29 years old), women, and the elderly has been rising steadily over the past few years. The city has been working on converting non-regular public sector workers to regular workers since 2015, and the rate dropped to 72.7% in 2019 (2015: 73.0%). As a result, the municipal administration and civil society must collaborate to raise the share of regular workers in the private sector and maintain the conversion of non-regular workers in the public sector to regular workers through consensus-building dialogues and administrative support. Furthermore, a substantial proportion of people work excessive hours, even though their wages and working conditions are only gradually improving. The government needs to find ways to reduce working hours and raise wages, especially for non-regular workers, as well as monitor industrial safety levels.

In a social economy, the city prioritizes people over capital (or profits), and the welfare of the entire society over the interests of businesses. This is becoming more important because of its obligations, such as generating services and providing jobs that are appropriate for the community's needs. Furthermore, through the Social Enterprise Support Center, the city of Suwon has been offering technical assistance to social companies, resulting in a continuous increase in the number of social enterprises. Additionally, Suwon has been running a specialist institution giving a variety of business help for start-ups, with an annual average of 12.6 start-ups from 2015 to 2019. SMEs received only KRW 1.09 billion in financial aid in 2019, so the city needs to evaluate its overall policies to encourage

sustainable business activities. The number of SMEs steadily increased to 120,000 in 2019 due to the transfer of funds to the Business Support Center for SMEs by the municipal government support of SMEs. Moreover, financial support for SMEs' technological growth has risen, particularly with the establishment of the Business Support Center for SMEs. It rose significantly to reach KRW 1.17 billion in 2019 and was used to improve SMEs' R&D environments, for the operation of shared research facilities, and for the promotion of relevant projects. It is important to strengthen financial support as technology advances and diversity increases. Suwon's six targets for quality jobs and industrial innovation are described in Table 5.

**Table 5.** Quality jobs and industrial innovation targets.

| No. | Target | Description | Indicator |
|---|---|---|---|
| E1 | Creating quality jobs | Increase the employment rate while decreasing the proportion of day and temporary workers. | 1. The employment rate (by different groups: aged 15 to 29, 45 to 60, the elderly, and people with disabilities). <br> 2. The fraction of regular workers among total paid workers. |
| E2 | Improvement of wages and working conditions | 1. Raise the wages of both temporary and regular workers in accordance with the principle of "equal pay for equal work". <br> 2. Decrease the weekly working hours to 32 h. <br> 3. Limit the number of industrial accidents per 10,000 people to 0.1. | 1. Weekly and annual working hours of wage workers (by gender and type). <br> 2. Wage workers' average earnings (by gender and kind of employment contract). <br> 3. The number of industrial accidents per 10,000 people in Suwon. |
| E3 | A good place to establish a business | 1. Establish a solid framework and infrastructure for all entrepreneurs, especially young individuals. <br> 2. Offer a variety of start-up assistance. <br> 3. Foster a climate in which start-ups can recover after a business failure. | 1. The number of firms that began with the support of the city. <br> 2. The rate of survival of start-ups passed through business incubation facilities. <br> 3. The amount of funding provided by the city to SMEs. |
| E4 | Completing the social economy ecosystem | 1. Restructure the governance framework in order to create a complete ecosystem for the social economy. <br> 2. Ensure that the social economy promotes local job development, social contributions, and innovation. | 1. The number of (preliminary) authorized social enterprises and cooperatives. <br> 2. The number of people working for the social economy. <br> 3. The speed of implementation of the city's Basic Plan for Social Economy. <br> 4. The proportion of vulnerable groups employed in the social economy. |
| E5 | Increasing the growth of small- and medium-sized businesses | 1. Create local infrastructure and a mechanism to assist in the innovation of SMEs. <br> 2. Extend benefits for SMEs in the public procurement market. <br> 3. Raise the budget allotment for developing skills in industries. | 1. The total number of SMEs. <br> 2. The level of growth in the budgets for entities that provide assistance to SMEs. <br> 3. The proportion of commodities purchased from SMEs in the public procurement market. |

| No. | Target | Description | Indicator |
|-----|--------|-------------|-----------|
| E6 | Establishing social infrastructure for industrial innovation in city | 1. Provide assistance to industries through policies, systems, and financing.<br>2. Emphasize urban infrastructure, scientific research, and industrial innovation for environmentally friendly technology industries. | The amount of the local government's annual financial support for industrial infrastructure and technological development research. |

### 4.1.6. Sustainable Consumption and Production

**SDGs of Responsible Consumption and Production**. The city of Suwon strives towards 'Sustainable Consumption and Production' to build a city where everyone lives in harmony with nature by boosting green production and consumption, minimizing general consumption and home waste, and raising people's knowledge of resource circulation. Green product production and consumption decrease environmental and societal expenses, such as ecological restoration costs. Purchasing green items has both social and economic benefits, according to customers. In addition, the city assesses green product purchases at public institutions to support the use of green products with a reduced environmental impact, lowering pollution while sustaining consumption and production. In 2019, the city's public institutions spent KRW 16.2 billion on green items, 56.26% of their general purchasing costs. Suwon has been promoting and enhancing environmental education since 2012, considering producer and consumer perspectives on sustainable consumption and production. Further, the city produced a visiting education service to show educational programs at recycling facilities, which increased from 43 in 2018 to 93 in 2019. Suwon's seven targets are listed in Table 6.

**Table 6.** Sustainable consumption and production targets.

| No. | Target | Description | Indicator |
|-----|--------|-------------|-----------|
| F1 | Promoting the purchase of environmentally friendly goods | 1. Double the quantity of eco-friendly green products acquired in the public procurement market<br>2. Continually increase the number of such products purchased each year to cover all commodities used by administrative agencies. | Rate of (eco-friendly) green product purchases. |
| F2 | Supporting companies with green certification | Provide incentives to local governments to encourage green companies and eco-friendly producers. | Number of companies with green certification. |
| F3 | Responsible consumption and waste reduction | Target higher waste reduction and recycling goals along with changes in business and consumer behaviors and producer activities. | Total waste generation and recycling rate. |
| F4 | Activating green markets | Increase green market. | The quantity of farmers and flea markets. |
| F5 | Promoting awareness of resource circulation | Develop sustainable consumption and production education in schools, media, and public institutions. | Number of eco-tours and possibilities for resource-circulation education. |
| F6 | Sustainable tourism | Develop and discover tourist infrastructure for cultural tourism and promote local jobs. | Budget allotted for promoting ethical travel and sustainable tourism. |

**Table 6.** *Cont.*

| No. | Target | Description | Indicator |
|-----|--------|-------------|-----------|
| F7 | Social economy with priority given to social values | 1. Ensure social values are met instead of lower prices in the public procurement market.<br>2. Every year, raise the proportion of products and services with social values in the city's procurement market. | The quantity of social economy purchases made in the city's public procurement market. |

4.1.7. Welfare, Health, and Education for Citizens' Happiness

**SDGs of No Poverty, Good Health and Well-being, and Quality Education.** The Social Security Master Plan is being implemented in the city of Suwon. With South Korea's relative poverty rate at 16.9% and the percentage of people receiving basic pensions at 60% in 2018, this suggests a comparatively high implementation rate. Given this, the city must enhance its programs to guarantee genuine financial stability and consistent quality of life after retirement. As a result, both the old poverty and the relative poverty rate will decrease. In 2017, the percentage of delinquents who were unable to pay insurance premiums due to difficult financial conditions rose to 33.33%. To reach its goal, the city needs assistance from the general public for new policy proposals as well as financial support for such offenders. Additionally, as a result of stricter welfare regulations, Suwon's population of 220,000 (1.8% of the total population) now receives National Basic Living Security.

To fulfill its goal of improving health services and preventative care, Suwon has established a local healthcare strategy. In 2019, 18.6% of Suwon's residents went to a public health facility, 22.5% engaged in moderate to vigorous physical exercise, and 78.4% had general health examinations. To further enhance the breadth and caliber of education, Suwon also offers opportunities to kids who are not enrolled in school and supports lifelong learning initiatives, libraries, and notably tiny community-based libraries. In 2019, the municipal budget increased for out-of-school youth to KRW 4.04 billion and for lifelong education to KRW 2.69 billion, and there was a drop in the total municipal budget of 0.11%. The spending plan for assisting libraries was cut to 15.51 billion (or 0.6% of the overall municipal budget). Small community-based libraries need a minimal administrative budget to be more accessible as part of the city plan. The targets are demonstrated in Table 7.

**Table 7.** Welfare, health, and education for citizens' happiness targets.

| No. | Target | Description | Indicator |
|-----|--------|-------------|-----------|
| G1 | Enhancing people's happiness levels | Create an index measuring citizens' happiness that evaluates their quality of life based on factors such as housing, environment, life expectancy, and health. | Citizens Happiness Index. |
| G2 | Ensuring minimum living conditions and reducing poverty | 1. Eliminate all problems with poverty by ensuring that everyone has a minimum standard of living.<br>2. Assist low-income populations in becoming self-sufficient.<br>3. Pay special attention to older residents when eradicating poverty in the city. | 1. The degree to which preparations for ensuring the bare necessities of life are being carried out.<br>2. Population in poverty percentage.<br>3. The proportion of older people who live in poverty.<br>4. The rate of public health insurance premium.<br>5. The percentage of National Basic Livelihood Security System beneficiaries in relation to the decile distribution ratio. |

**Table 7.** *Cont.*

| No. | Target | Description | Indicator |
|---|---|---|---|
| G3 | Promoting health services and preventive care | Provide physical and mental health services for low-income and vulnerable groups, including young women, children, disabled people, and the elderly. | 1. The frequency of visitors to medical facilities.<br>2. The amount of exercise performed by people with severe problems.<br>3. Self-reported state of health.<br>4. The proportion of public sporting venues per 10,000 residents.<br>5. The number of persons who have routine health exams.<br>6. Mortality rate from suicide. |
| G4 | Application of universal design | Embrace universal design to build a truly inclusive city with happy elderly residents, disabled residents, children, and women that ensures mobility rights for all. | City Universal Design Index. |
| G5 | Improving the quality of universal mandatory education | 1. Offer a variety of educational programs and regulations to raise the standard of universally required education.<br>2. Establish a range of alternative educational institutions and initiatives to educate teenagers who are not enrolled in school. | 1. The quantity of instances in primary and secondary schools where the average number of students in a class is exceeded.<br>2. The number of cognitive tests offered to parents of pupils.<br>3. The proportion of teenagers not in school.<br>4. The funds allocated by the city to support young people who are not in school. |
| G6 | Encouraging lifelong learning and democratic citizenship education | 1. Include courses for lifelong learning as well as public education programs' curricula on democratic citizenship and human rights.<br>2. Establish educational institutions, programs, and funding to support lifelong learning in humanities, physical education, social culture, and democratic citizenship. | 1. The Provincial Office of Education's Democracy Index of Schools in the city.<br>2. The annual spending allotted for education.<br>3. The proportion of people taking part in lifelong learning initiatives. |
| G7 | Diversify, promote, and enhance access to libraries | Diversify library content, increase infrastructure and budget, and provide citizens with cultural and educational resources. | 1. Amount of budget for libraries.<br>2. Trends in budget for supporting small libraries |

### 4.1.8. Gender, Equality, and Multicultural Society

**SDGs of Gender Equality.** The city of Suwon has developed long-term educational programs for gender equality and balance in policymaking. Discrimination based on gender, religion, place of origin, or culture must be eliminated, and sexual and domestic violence must be reduced. It is critical to respect cultural differences and increase cultural awareness, not only to minimize disputes caused by misconceptions but also to build a society in which everyone has the freedom to pursue happiness. In 2019, 34% of the city of Suwon's public officials completed gender equality education programs. Since 2015, the annual number of gender equality education classes in elementary, middle, and high schools has dropped, with only 36 classes recorded in 2018. In 2019, this number climbed to 52 classes. Furthermore, the gender balance in leadership roles in Suwon's

governmental offices has steadily improved, with male officials holding 84% of leadership positions. Given the increasing number of newly recruited female public officials, this gender imbalance is likely to shrink. Furthermore, the year-on-year growth in the number of instances involving sentenced offenders was 16.4% (2163 cases), indicating that the municipal government should re-evaluate the overall system. The city of Suwon also formed a department committed to fostering multiculturalism in which the budget has gradually expanded. Suwon established the five targets listed in Table 8.

**Table 8.** Gender, equality, and multicultural society targets.

| No. | Target | Description | Indicator |
|---|---|---|---|
| H1 | Strengthening the quality of education on gender equality | 1. Every year, perform a quantitative and qualitative assessment of gender equality education. 2. Improve teenager education on the topic of gender equality and human rights. | 1. Budget for gender equality and human rights education. 2. The proportion of public institutions that provided gender equality education. 3. The number of opportunities for teenagers to learn about gender equality. |
| H2 | Building governance for gender equality | Create a gender equality monitoring team and promote gender equality for seniors, disabled persons, children, and migrants. | 1. The number of gender equality monitoring activities. 2. The percentage of women serving as chairperson or vice-chair of various administrative committees. 3. Gender ratio of public officers in grade five or higher. |
| H3 | Putting an end to sexual violence | Release reports on sexual assault cases, domestic violence, sex trafficking victims, and human rights violations, identify necessary reforms, and implement them into policy. | 1. The total number of sessions for counseling made available to victims of domestic abuse, sexual violence, dating violence, and workplace sexual harassment. 2. The budget allocated to victims of sexual violence. |
| H4 | Guaranteeing minimum living standards for foreign residents in city | Establish a basic social security program to assist foreign residents in meeting their fundamental needs (safety, housing, education, and health care). | Quantity of the city budget allotted to the administrative staff in charge of multiculturalism promotion. |
| H5 | Improving citizens' awareness of multiculturalism | Build a welcoming social environment to increase individuals' acceptance and awareness of multiculturalism. | 1. The number of people who participate in educational programs offered by the city's primary and secondary schools. 2. The total number of people who participate in educational programs offered by the city's public institutions. |

4.1.9. Sustainable City and Culture

**SDGs of Sustainable Cities and Communities**. The city of Suwon must develop and manage its infrastructure to be acceptable and sustainable for all of its citizens; hence, the city's population continues to expand slowly. It avoids putting a strain on existing infrastructure and alleviates population concentrations in specific locations. Currently, there are enough residences in Suwon to accommodate 100% of the population. On the

other hand, the provision of cheaper and pleasant public housing for low-income families who are dissatisfied with their current surroundings has increased. In particular, the city has just started a pilot housing initiative to build and manage public rental homes. A city housing corporation and a more secure residential area have been suggested as new approaches. To increase the number of opportunities for citizens to engage in cultural activities, it has also increased the budget for cultural projects and the number of cultural venues. However, the amount of people visiting cultural or athletic venues has not kept up with Suwon's population development. Suwon's plan to become a sustainable cultural city includes the five targets listed in Table 9.

**Table 9.** Sustainable city and culture targets.

| No. | Target | Description | Indicator |
|---|---|---|---|
| I1 | Governing the city based on the capability of the urban environment and assisting inhabitants in receiving essential services | 1. Providing a high quality of life by integrating multiple elements (density, size, air quality, accessibility, and ecological environment). 2. Performing assessments of urban environment capacity. 3. Assisting citizens in receiving essential services through urban management. | 1. Population density. 2. Population decrease rate by dong (administrative unit). 3. The rate at which the number of firms shrinks by dong (administrative unit). 4. The degree to which buildings have aged. 5. Park area per capita of the city. |
| I2 | Ensuring individuals' housing rights and developing public rental housing | 1. Protect citizens' right to housing, develop city rules on public rental housing, and protect people's living conditions. 2. Increase the number of public housing units for one- and two-person households while tackling diverse areas, including the original city core. | 1. The fraction of public rental housing to the overall housing stock. 2. The percentage of public rental housing in the annual housing supply. |
| I3 | Mitigating the burden of housing expenses and promoting social housing | Increase welfare policies for the provision of social housing, including youth dormitories, housing cooperatives, and rental housing for low-income households. | 1. The ratio of housing rental fees to citizens' income. 2. The share of tenant households in relation to total households. |
| I4 | Ensuring citizens' access to cultural resources through sustainable cultural policies | Create a public–private governance committee for sustainable cultural policy by launching and operating the Culture and Arts Committee to guarantee the citizens' access to culture. | 1. Cultural budget: Amounts of total and sectoral budget. 2. The number of cases of supporting artists and cultural groups. 3. The number of cultural infrastructure assets per 100,000 people. |
| I5 | Cultural rejuvenation and enrichment for citizens | Encourage people's cultural enjoyment and encourage citizens to become active participants in cultural and arts activities as both consumers and producers of culture. | 1. The number of cultural and arts exhibitions. 2. The number of people who visit cultural institutions. 3. The quantity of cultural and arts clubs and their membership. |

#### 4.1.10. Autonomy, Justice, Peace, and Harmony through Self-Governance

**SDGs of Peace, Justice, and Strong Institutions.** The city of Suwon has been releasing local government information to the public and assisting individuals in developing their role in participatory governance. Furthermore, the city created procedures for measuring corruption and promoting public office integrity. These initiatives are recommended to increase citizens' trust in government, which serves as the foundation for participatory governance. Simultaneously, there has been a continuous debate about the criteria for establishing indicators of administrative units' self-governance and citizens' capacity for participatory governance. In addition, as part of its information disclosure system, the city has devised a system for requesting administrative information from local governments. Without being requested, the city has made original records signed by the deputy mayor or higher available to the public. Aside from that, the number of committees, which serve as representative forms of citizens' self-government, has continuously expanded since 2015. The gender ratio of commissioned members has been maintained without prejudice towards one side from 2009 to 2019, with 179 active committees. Furthermore, the city has been designated as an exemplary city after receiving a score of 2 or better in the Anti-Corruption Initiative Assessment (AIA), which is meant to safeguard the integrity of public officials. Suwon established the six goals listed in Table 10.

**Table 10.** Autonomy, justice, peace, and harmony through self-governance targets.

| No. | Target | Description | Indicator |
|---|---|---|---|
| J1 | Strengthening self-governance capacity by dong (administrative unit) | Increase the budget to strengthen self-governance and establish yearly plans for residents' associations. | The number of participants in educational programs on capacity building for self-governance of residents' associations by dong. |
| J2 | Guaranteeing access to administrative information | Enhance transparency and accountability by making the minutes of city committee meetings public to provide citizens with access to information. | 1. Rate of the full textual disclosure of documents produced and received by the administration. 2. The number of cases of textual disclosure of the minutes of the city's committee meetings. |
| J3 | Inclusive governance for all | 1. Encourage citizens' participation in government to promote participatory democracy, such as having residents participate in village budget projects or assembling a group of citizens to plan a policy. 2. Involve minorities and the vulnerable in various governing organizations and committees to build a diverse decision-making structure. | 1. The proportion of cases in which stakeholders were consulted during the planning stages of official plans, important basic plans, and key projects. 2. The portion of the yearly budget allocation based on citizen engagement. 3. The number of policies presented by civic governance organizations that have been adopted (by category and industry). 4. The percentage of women, people with disabilities, the elderly, and teenagers who participate in civic government organizations. 5. Municipal government's credibility for policy decision making. |
| J4 | Ensuring public officials' transparency and honesty | Preserve the administration's transparency and integrity and seek to be a city free of corruption by public officials and firms providing bribes, with the media and civil society monitoring corruption. | 1. The annual number of corruption cases involving public officials. 2. The Integrity of Public Officials Index. |

**Table 10.** *Cont.*

| No. | Target | Description | Indicator |
|---|---|---|---|
| J5 | Build a safe city for everyone | 1. Establish a tranquil and safe city, even at night. 2. Minimize all types of violence and violence-related mortality; and enforce the law and provide equal access to the court system. | 1. The number of violent crimes. 2. The number of crime-related deaths per 100,000 people. |
| J6 | Promoting the understanding of democratic citizenship and human rights in administration, businesses, and the public | Boost city administration's and businesses' human rights knowledge, and encourage democratic citizenship among citizens. | 1. The number of human rights education opportunities for citizens and public officials. 2. The proportion of the city's budget allocated to democratic citizenship education. |

### 4.2. Smart Sustainable City Indicators

The indicators developed by [4] attempt to include more non-technological aspects along with covering sustainable goals, categorized into four different aspects (socio-cultural, economic, environmental, and governance). They also bring the concept of sustainable development into the indicators. For these two reasons, the indicators [4] are thoroughly investigated along with the contextual indicators developed by the city of Suwon, which are called SSDGs and are described in terms of 55 sub-indicators as shown in Table 11.

**Table 11.** Smart Sustainable City Indicators.

| No. | Category | Indicator | Sub-Indicators |
|---|---|---|---|
| K1 | Socio-Cultural | Healthcare delivery | • Percentage of population with access to primary health care facilities (SS01-01) • The number of people with the immunizations against infectious childhood diseases (SS01-02) • The contraceptive prevalence rate (SS01-03) |
| | | Quality drinking water | The number of populations with access to safe drinking water (SS02-04) |
| | | Individuals' health monitoring | • The number of services integrated in a singular operation center leveraging real-time data. 1 point for each: ambulance, emergency/disaster response, fire, police, weather, transit, and air quality (SS03-05) • The percentage of residents live individual, integrating health historical data facilitating patient and health provider to access the complete medical records (SS03-06) |
| | | Quality food | • The number of children with a poor nutritional status (SS04-07) • The nutritional status of the population (SS04-08) |
| | | Education Funding | The number of services and resources that receive education funding (SS05-09) |
| | | Free Education | • The number of children reaching grade five of primary school (SS06-10) • The level of achievement for adult secondary education (SS06-11) |
| | | Low crime rate | The violent crime rate per 100,000 people (SS07-12) |
| | | Population density | Population-weighted density (the average densities of separate census tracts) (SS08-13) |
| | | Population growth rate | The population growth rate (SS09-14) |
| | | Investment in culture | The percentage of municipality budget that is allocated to culture-related programs (SS10-15) |
| | | Civic engagement | • The number of civic engagement activities held by the municipality in the previous year (SS11-16) • Voting in previous municipal election (as a percentage) (SS11-17) |

**Table 11.** *Cont.*

| No. | Category | Indicator | Sub-Indicators |
|---|---|---|---|
| K2 | Economic | Affordable housing | The percentage of inhabitants with housing deficiency in any of the following five areas: potable water, sanitation, overcrowding, deficient material quality, or lack of electricity (SS12-18) |
| | | Start-ups | The number of new start-ups annually (SS13-19) |
| | | International collaboration | The number of international attendees of conferences and events (SS14-20) |
| | | Low poverty rate | The poverty rate (SS15-21) |
| | | Job opportunities | • The employment rate (SS16-22)<br>• The percentage of labor force (LF) engaged in creative industries (SS16-23) |
| K3 | Environmental | Green spaces | The area of green spaces per 100,000 m2 (SS17-24) |
| | | Air quality | The concentration of air pollutants in urban areas (SS18-25) |
| | | Low pollution | Measurement of particulate matter (PM2.5 and PM10), ozone (O3), nitrogen dioxide (NO2), sulfur dioxide (SO2), and carbon monoxide (CO) emissions (SS19-26) |
| | | Energy use | • The energy consumption per capita per year (SS20-27)<br>• The consumption rate of renewable energy resources (SS20-28)<br>• The energy use intensity (SS20-29) |
| | | Waste production | • The amount of industrial and municipal solid waste (SS21-30)<br>• The amount of hazardous waste generated (SS21-31)<br>• The amount of radioactive waste generated (SS21-32)<br>• The amount of waste recycling and reuse (SS21-33) |
| | | Sustainability-certified buildings | • The number of LEED or BREAM sustainability-certified buildings in the city (SS22-34)<br>• The percentage of commercial and industrial buildings with smart meters (SS22-35)<br>• The percent of commercial buildings with a building automation system (SS22-36)<br>• The percent of homes (multifamily and single family) with smart meters (SS22-37) |
| K4 | Governance | E-governance | • The use of open data (SS23-38)<br>• The number of mobile apps available (iPhone) based on open data (SS23-39)<br>• The existence of official regulation on privacy to protect confidential citizen data (SS23-40) |
| | | Real-time data monitoring | • The presence of demand-based pricing (e.g., congestion pricing, priced toll lanes, priced parking spaces). (SS24-41)<br>• The percentage of traffic lights connected to real-time traffic management system. (SS24-42)<br>• The number of public transit services with real-time information for the public: buses, regional trains, metros, rapid transit systems (e.g., BRT, tram), and sharing modes (e.g., bike sharing, car sharing) (SS24-43)<br>• The availability of multimodal transit application with at least three services integrated (SS24-44) |
| | | Internet and Wi-Fi coverage | • The number of internet subscribers per 1000 inhabitants (SS25-45)<br>• The percentage of commercial and residential users with internet (download) speeds of at least 2 Mbit/s (SS25-46)<br>• The percentage of users (commercial and residential) with internet download speeds of a minimum of 1 gigabit/s (SS25-47) |
| | | Disaster preparedness | The economic impact and the number of human losses due to natural disasters (SS26-48) |
| | | Public transport | • The number of public transport trips per capita per year (SS27-49)<br>• The percentage of non-motorized transport trips among total transport trips (SS27-50)<br>• The integrated fare system for public transport (SS27-51) |
| | | Clean energy transportation | • The length (in kilometers) of bicycle paths and lanes per 100,000 residents (ISO 37120: 18.7) (SS28-52)<br>• The percentage of shared bicycles per capita (SS28-53)<br>• The number of shared vehicles per capita (SS28-54)<br>• The number of electric vehicle charging stations in the city (SS28-55) |

### 4.3. Content Analysis

**Content Selection.** The content for the analysis was taken from the SSDGs and SSCIs. There are 10 groups that were originally designed from the SDGs as manifested in 126 indicators under the SSDGs, and there are 55 indicators from the SSCIs. The analysis uses these indicators.

**Text Preprocessing.** This study utilizes common text processing, such as tokenization, stemming, and the removal of stop words. Tokenization separates each word or symbol into a token. Stemming (i.e., Porter stemming) looks for the root of the word. Subsequently, the removal of stop words is used to retain important words and remove frequently occurring words such as "a", "an", "the", "on", etc. The preprocessed data were transformed into vectors with TF/IDF. The TF/IDF vector of each (both SSDGs and SSCIs) is measured using cosine similarity. The similarity matrix is then utilized to look for the commonality of the indicators between the two sets.

**Filter.** The overview of the similarity (i.e., the similarity of descriptions between pairs of indicators) can be seen in Figure 7. The meaning of similarity between a pair of indicators is that one of the indicators of SSDGs has a high similarity value with one of the indicators of SSCIs.

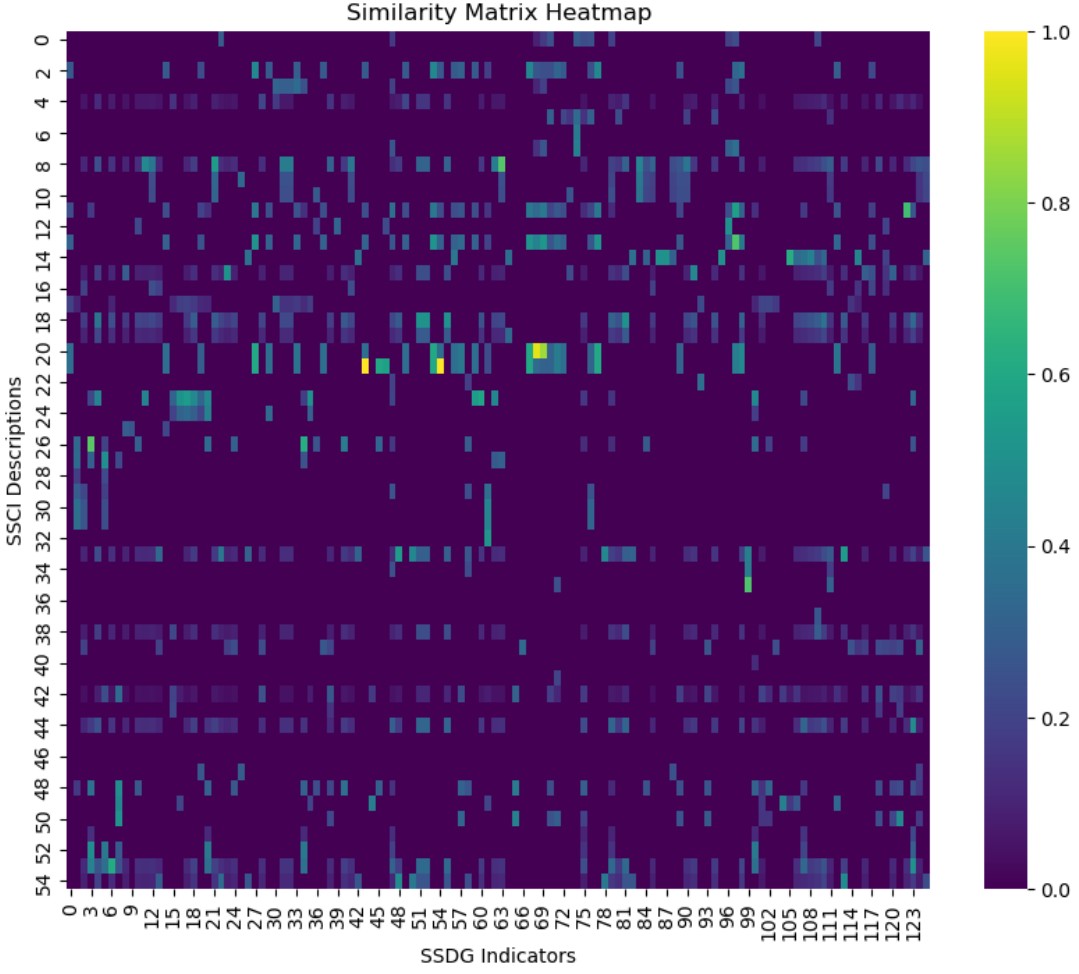

**Figure 7.** Heatmap of the similarity value between SSDGs and SSCIs.

The numbers in rows and columns refer to the index of the indicators in the SSCIs and SSDGs, respectively. Since the result is sparse (i.e., there are many zeros), filtering could enhance the readability of similar indicators. By setting the threshold to 0.5, there are 48 relevant pairs of indicators extracted. However, some pairs have inconsistent semantics. For example, the similarity value is 1.0 for SS16-22 (the employment rate) and

both 5-1-1 (i.e., the employment rate by different groups) and E4-4 (the proportion of vulnerable groups employed in the social economy). While the semantic similarity of SS16-22 and E1-1 is somehow correlated, the semantic similarity of SS16-22 and E4-4 is different. Another example is SS05-09, which has high similarity values with B4-1 (0.548) and 6-5 (0.732). Looking into the details of the indicators, they have similar keywords on education. Meanwhile, the semantics are different, since indicator SS05-09 deals with the number of services and resources for education funding while indicator B4-1 refers to the number of schools with an educational program on biodiversity, and indicator F5 considers the number of eco-tours and possibilities for resource circulation education. Due to this issue, coding and another round of manual investigation among the pairs of indicators are necessary.

**Coding.** Among the 48 pairs of indicators, we manually investigate the relevant similar indicators. The reason for performing this manual investigation is to identify the potential relationships among pairs of indicators. A manual investigation consists of two steps. The first step is to analyze the pairs of indicators' descriptions. Subsequently, the second step is to look for the semantic relationships among different possible pairs of indicators.

**Analysis.** The analysis step of the first step looks for the potential relationships of the indicators to the three pillars (i.e., economic, environment, and social). The analysis includes an additional aspect in the indicators: governance. Sixteen out of the fifty-five SSCI indicators listed are similar to the indicators of SSDGs: 6 out of 17 of the socio-cultural indicators (35.3%), 3 out of 6 of the economic indicators (50%), 5 out of 14 of the environmental indicators (35.7%), and 2 out of 18 of the governance indicators (11%).

The domination of the similarity of the pairs of indicators was shown in terms of the economic aspect by 17 pairs of indicators, followed by the socio-cultural aspect by 16 pairs of indicators, the environment aspect by 11 pairs of indicators, and the governance aspect by 3 pairs of indicators.

After the second step, the pairs of indicators can be analyzed as belonging to one of four groups: similar-recognized, dissimilar-recognized, similar-unrecognized, and dissimilar-unrecognized. Similar-recognized refers to the descriptions that are the same and semantically similar (e.g., the pairs between SS16-22 and E1-1). *Dissimilar-recognized* indicates that it is recognized as similar while it is dissimilar (e.g., the pairs between SS16-22 and E4-4). *Similar-unrecognized* is difficult to trace using the similarity value since the descriptions might be different while the point is the same (e.g., the pairs between SS19-26 and A4-3, as well as SS18-25 and A4-3). The remaining indicators are considered *dissimilar-unrecognized*, and these are not analyzed. Hence, implicit descriptions of indicators cause some challenges in recognizing similarities.

In the second step, we found that there are some additional indicators that are similar in the pairs, but without much relevancy. In this case, we focus on dissimilar-recognized, similar-unrecognized, and dissimilar-unrecognized. We manually investigate the indicators and analyze them accordingly in two aspects: an ambiguous analysis and an unrelated analysis. The ambiguous analysis refers to multiple interpretations and lacks specificity for the indicators. This ambiguity can arise from various factors, including incomplete words, vague language, or a lack of context. The unrelated analysis involves indicators (SSDGs) that do not have a direct or relevant connection to the other indicators (SSCIs). The ambiguous analysis and the unrelated analysis can be seen in Tables 12 and 13, respectively. In this study, we keep the indicators in the ambiguous analysis and remove the indicators based on the unrelated analysis.

After considering the second step, 19 out of the 55 SSCI indicators (34.5%) are listed as the indicators of SSDGs: 7 out of 17 of socio-culture indicators (41.1%), 4 out of 6 of economic indicators (66.6%), 6 out of 14 of environment indicators (42.9%), and 3 out of 18 of governance indicators (16%). The domination of the similarity of the pairs of indicators was shown in terms of the economic aspect by 11 pairs of indicators, followed

by the socio-cultural aspect by 7 pairs of indicators, the environment aspect by 7 pairs of indicators, and the governance aspect by 2 pairs of indicators.

**Table 12.** Ambiguous analysis of the pairs of indicators.

| Pairs of Indicators | Description |
|---|---|
| SS05-09 and B4-1 | It is related to educational funds. Meanwhile, B4-1 is specific for biodiversity |
| SS11-16 and B5-1 | It is related to civic engagement activities. Meanwhile, B2-1 is specific for biodiversity |
| SS13-19 and E4-1 | It is about the number of start-ups with new opportunities while E4-1 is about social enterprises. |
| SS13-19 and E4-2 | It is about the number of start-ups with new opportunities while E4-2 is about the number of people who are working for the social economy |

**Table 13.** Unrelated analysis of the pairs of indicators.

| Pairs of Indicators | Description |
|---|---|
| SS09-14 and C1-3 | SS09-14 is about population while C1-3 is about the growth of biodiversity |
| SS15-21 and C1-3 | SS15-21 is about the poverty rate while C1-3 is about the growth of biodiversity |
| SS16-22 and C1-3 | SS16-22 is about the employment rate while C1-3 is about the growth of biodiversity |
| SS20-27 and C4-2 | SS20-27 is about annual energy consumption while C4-2 is about water consumption |
| SS22-34 and E3-1 | SS22-34 is about sustainability-certified buildings in the city while the E3-1 is about the number of support of the city |
| SS09-14 and E4-3 | SS09-14 is about the population growth rate while E4-3 is about the growth rate of the basic plan for the social economy |
| SS15-21 and E4-3 | SS15-21 is about the poverty rate while E4-3 is about the growth rate of the basic plan for the social economy |
| SS17-24 and F1 | SS17-24 is about the green area per 100,000 $m^2$ while F1 is about the rate of green product purchases |
| SS17-24 and F2 | SS17-24 is about the green area per 100,000 $m^2$ while F2 is about the number of companies with green certification |
| SS09-14 and G2-1 | SS09-14 is about the population growth rate while G2-1 is about the rate of implemented plans for guaranteeing the minimum living standards |
| SS15-21 and G2-1 | SS15-21 is about the poverty rate while G2-1 is about the rate of implemented plans to guarantee the minimum living standards |
| SS16-22 and G2-1 | SS16-22 is about the employment rate while G2-1 is about the rate of implemented plans for guaranteeing the minimum living standards |
| SS09-14 and G2-3 | SS09-14 is about the population growth rate while G2-3 is about the elderly population and elderly poverty |
| SS09-14 and G3-1 | SS09-14 is about the population growth rate while G3-6 is about the suicide mortality rate |
| SS15-21 and G2-1 | SS15-21 is about the poverty rate while G3-6 is about the suicide mortality rate |
| SS16-22 and G2-1 | SS16-22 is about the employment rate while G3-6 is about the suicide mortality rate |
| SS13-19 and G5-3 | SS13-19 is about the number of new-opportunity-based start-ups while G5-3 is about the number of out-of-school teenagers |
| SS10-15 and H3-2 | SS10-15 is about the percentage of the municipal budget for culture while H3-2 is the amount of the budget for sexual violence victims |
| SS07-12 and I1-2 | SS07-12 is about the violent crime rate per 100,000 people while I1-2 is about the population decrease rate by the administrative unit |
| SS09-14 and I1-2 | SS09-14 is about the population growth rate while I1-2 is about the population decrease rate by the administrative unit |
| SS22-36 and I1-4 | SS22-36 is about the percentage of commercial buildings with a building automation system while I1-4 is about the degree of aging of buildings |
| SS22-34 and J2-2 | SS22-34 is about the number of LEED or BREAM sustainability-certified buildings in the city while J2-2 is about the number of cases of textual disclosure of the minutes of the city of Suwon's committee meetings |

## 5. Discussion and Conclusions

This study proposed an approach to investigate the concept of smart sustainable city indicators with the city sustainable development indicators developed through participatory processes. The proposed approach utilized a content analysis along with text analytics to see the alignment between the two concepts: SSDGs and SSCIs. The SSDG indicators, such as policies, public services, city management and administration, the knowledge

economy, governance, participation and collaboration, the built environment and city infrastructure, human capital and creativity, information and communication technologies (ICT), the natural environment and ecological sustainability, and data and information, were all considered in this content analysis. This study found that the concept of SSCIs requires further investigation to comply with the sustainable development goals (SDGs).

Based on the analysis, the SSDGs comply with 34.5% of the SSCIs. Most of the pairs of indicators comprise the economic (66.6%), environment (42.9%), socio-cultural (41.1%), and governance (16%) dimensions, respectively. While the two main characteristics of Suwon's policy are based on its environment and socio-culture, the dominant aligned indicators are economics. Suwon's citizens' participation is unnoticeable and the promotion of biodiversity is barely perceivable in the SSCIs.

### 5.1. Practical Implications

Suwon stands as an example of South Korea's smart city initiatives. Suwon's current Sustainable Development Goals (SSDGs) need to be assessed for the harmonization between SSDGs and SSCIs. Our primary aim is to disseminate the city of Suwon's present status and advancements, offering valuable insights for other municipalities as they adopt their individualized Sustainable Development Goals tailored to their specific contextual realities. Furthermore, our analysis delves into the alignment between the SSDGs and the SSCIs, enriching our understanding of their synergies.

This outcome has the potential to encourage other municipalities to prioritize the importance of SSCIs in conjunction with the Sustainable Development Goals (SDGs) while formulating their contextual urban sustainability strategies. By exclusively relying on smart city indicators, certain essential sustainable benchmarks might be difficult to attain. Consequently, this underscores the necessity of a comprehensive approach that embraces both domains to ensure an effective smart city development framework. As a comparison, a previous case study of the Municipality of Lisbon [48] focused on one aspect of policy making for future cities, air quality, using data science. Therefore, they did not examine the SDGs as a whole in their analysis (Table 14).

### 5.2. Limitations and Future Research

The strength of this analysis lies in its elaboration as explained previously in the results section. There have been no previous similar research studies; hence, it is hoped that this research might contribute a new perspective on conducting an evaluation of sustainable smart city policies implemented in various cities in the world. However, there are some limitations that should be addressed. First, the use of text analytics may have hindered our search for similar indicators due to the culture and geographical context. In the cultural context, the indicators of the city of Suwon are based on a people-friendly city with two main policies: socio-culture and biodiversity. Although the SSDG indicators adopted the SDGs, Suwon's policies rely on the indicators in the local context, such as the employment rates for every age category. In terms of the geographical context, some geographical structures contain indicators, such as streams. Meanwhile, the SSCIs are a conceptual framework that attempts to generalize smart city indicators. These limitations require manual investigation, which might be challenging for researchers in future case studies.

Therefore, a basic text analysis is recommended as a prior step to prevent unnecessary investigations. More advanced semantic analyses, such as geospatial analyses and contextual semantic search (CSS), might be adopted in future studies. In addition, the enabling of a cross-functional country among the sister cities could enhance investigations into smart sustainable city development in the long run. Data-driven smart sustainable city development plans would also provide further insight into indicators for municipalities.

**Table 14.** Similarity values are based on the pairs of indicators (similar-recognized (black), similar-unrecognized (blue and red), manually investigation (xxx)). The green font indicates the unrelated analysis.

| No. | SS02-04 | SS05-09 | SS07-12 | SS08-13 | SS09-14 | SS10-15 | SS11-16 | SS13-19 | SS15-21 | SS16-22 | SS16-23 | SS17-24 | SS18-25 | SS19-26 | SS20-27 | SS21-33 | SS22-34 | SS22-36 | SS27-51 | SS28-54 |
|---|---|---|---|---|---|---|---|---|---|---|---|---|---|---|---|---|---|---|---|---|
| A2-1 | | | | | | | | | | | | | | | 0.74 | | | | | |
| A4-1 | | | | | | | | | | | | | | | | | | | | 0.61 |
| A4-2 | | | | | | | | | | | | | | | | | | | | 0.51 |
| A4-3 | | | | | | | | | | | | | xxx | xxx | | | | | | |
| B2-1 | | | | | | | | | | | | 0.51 | | | | | | | | |
| B2-2 | | | | | | | | | | | | 0.54 | | | | | | | | |
| B4-1 | | 0.54 | | | | | | | | | | | | | | | | | | |
| B5-1 | | | | | | | 0.50 | | | | | | | | | | | | | |
| C1-3 | | | | | 0.51 | | | | 0.60 | 0.60 | | | | | | | | | | |
| C4-2 | xxx | | | | | | | | | | | | | 0.62 | | | | | | |
| C5-1 | | | | | | | | | | | | 0.51 | | | | | | | | |
| E1-1 | | | | | | | | | | 1.0 | | | | | | | | | | |
| E2-1 | | | | | | | | | | 0.59 | | | | | | | | | | |
| E2-2 | | | | | | | | | | 0.57 | | | | | | | | | | |
| E3-1 | | | | | | | | | | | | | | | | | | 0.55 | | |
| E4-1 | | | | | | | | 0.51 | | | | | | | | | | | | |
| E4-2 | | | | | | | | 0.51 | | | xxx | | | | | | | | | |
| E4-3 | | | | | 0.51 | | | | 0.60 | | | | | | | | | | | |
| E4-4 | | | | | | | | | | 1.0 | | | | | | | | | | |
| E5-1 | | | | | | | | 0.51 | | | | | | | | | | | | |

**Table 14.** *Cont.*

| No. | SS02-04 | SS05-09 | SS07-12 | SS08-13 | SS09-14 | SS10-15 | SS11-16 | SS13-19 | SS15-21 | SS16-22 | SS16-23 | SS17-24 | SS18-25 | SS19-26 | SS20-27 | SS21-33 | SS22-34 | SS22-36 | SS27-51 | SS28-54 |
|---|---|---|---|---|---|---|---|---|---|---|---|---|---|---|---|---|---|---|---|---|
| F1 | | | | | | | | | | | | 0.55 | | | | | | | | |
| F2 | | | | | | | | | | | | 0.59 | | | | | | | | |
| F3 | | | | | | | | | | | | | | | | | | 0.52 | | |
| F5 | | 0.73 | | | | | | | | | | | | | | | | | | |
| G2-1 | | | | | 0.51 | | | | 0.60 | 0.60 | | | | | | | | | | |
| G2-2 | | | | | | | | | 0.96 | | | | | | | | | | | |
| G2-3 | | | | | 0.53 | | | | 0.86 | | | | | | | | | | | |
| G3-6 | | | | | 0.51 | | | | 0.60 | 0.60 | | | | | | | | | | |
| G5-3 | | | | | | | | | 0.51 | | | | | | | | | | | |
| G5-4 | | xxx | | | | | | | | | | | | | | | | | | |
| G6-2 | | xxx | | | | | | | | | | | | | | | | | | |
| G7-1 | | | | | | 0.5 | | | | | | | | | | | | | | |
| G7-2 | | | | | | 0.5 | | | | | | | | | | | | | | |
| H3-2 | | | | | | 0.5 | | | | | | | | | | | | | | |
| I1-1 | | | | 0.51 | | | | | | | | | | | | | | | | |
| I1-2 | | | 0.56 | | 0.73 | | | | | | | | | | | | | | | |
| I1-4 | | | | | | | | | | | | | | | | | | | 0.72 | |
| I4-1 | | | | | | 0.62 | | | | | | | | | | | | | | |
| J2-2 | | | | | | | | | | | | | | | | | | 0.55 | | |
| J5-1 | | 0.69 | | | | | | | | | | | | | | | | | | |
| J5-2 | 0.51 | | | | | | | | | | | | | | | | | | | 0.51 |

**Author Contributions:** Conceptualization, R.J.P., A.K.K., B.N.Y. and Y.A.I.; Methodology, R.J.P. and B.N.Y.; Writing—original draft, R.J.P. and A.K.K.; Writing—review and editing, R.J.P., A.K.K., Y.A.I. and B.N.Y. All authors have read and agreed to the published version of the manuscript.

**Funding:** This research received no external funding.

**Institutional Review Board Statement:** Not applicable.

**Informed Consent Statement:** Not applicable.

**Data Availability Statement:** Not applicable.

**Acknowledgments:** We would like to thank Thomhert Suprapto Siadari, from the Association of Indonesian Researchers in Korea (APIK) 2022–2024, APIK Smart City Task Force, and The Embassy of the Republic of Indonesia in Seoul for the ideation and constructive suggestions in preparing this manuscript.

**Conflicts of Interest:** The authors declare no conflict of interest.

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
