# Peer review of "Investigation of Smart Sustainable City Indicators of Sustainable Development—A Case Study of the City of Suwon"

_sustainability, doi:10.3390/su151914283_

Round 1

Reviewer 1 Report

The article entitled “Investigation of Smart Sustainable City Indicators on Sustainable Development – a case study on Suwon City" proposed a good topic; however, based on my review, i would like to rise up some comments, questions and recommendations.

- Several references can be merged like [4], [5], [7], [8], [9], [10] and [11]– [13].

- Related similar studies need to be added to clarify in a table the research gap. In addition, the structure of the manuscript can be figured besides the text.  

- Discussion:

The authors need to discuss the results through the following points.

i. Comparison with other studies

ii. Implication and explanation of findings

iii. Strengths and limitations

iv. Conclusion, recommendation, and future direction.

-  Conclusion:

The authors need to revise this part and highlight conducted results.

I recommend giving second chance to the author to check the grammar and scientific language carefully.

Author Response

The response to reviewer can be seen in the PDF file. Thank you.

Reviewer 2 Report

Very well organised paper, with concrete methodological approach, rich and up to date literature review and clear results.

Some comments that would improve the paper’s quality are as follows :

1. Methodological issues : Content analysis is not sufficiently described. More particularly, the author(s) mention that “The contents for the analysis were taken from SSDG and SSCI. There are 10 groups designed from SDGs with 126 indicators from the SSDGs…”

Moreover, these 126 indicators were developed by the city’s authorities in accordance to the existing Global indicator framework for the Sustainable Development Goals ? (https://unstats.un.org/sdgs/indicators/Global%20Indicator%20Framework%20after%202023%20refinement_Eng.pdf). This has to be explained because it is not clear to the reader. If that is the case, why didn’t the authors used the SDGs indicators directly ? Please clarify...

In line 539, page 21, the authors mention “The 55 indicators are shown in Table 11". They are actually sub-indicators (the indicators are included in the previous column). This has to be checked in the whole text.

2. In general all the text requires some careful proofreading before final submission..

I.e “Suwon City has adopted 10 Sustainable Development Goals (SDGs) for each sustainability ..”

this sentence needs rephrasing

3. In addition, since the titles of the 10 paragraphs regarding to these goals are not the same with the SDGs original titles , at the beginning of each paragraph it must be mentioned to which specific SDG each paragraph refers.

4. In page 5 , line 174... “....Other work gave a thorough assessment of the literature on smart sustainable cities 174 and discussed the benefits and challenges of developing such cities. The study identified …” reference is missing…

careful proof reading is requierd before tha final submission 

Author Response

(The authors gave the same response as above.)

Reviewer 3 Report

This paper analyses the relationship between city sustainability development indicators and smart sustainable city indicators (SSCI). To do that, Suwon City is considered as a case study. Suwon City sustainable development policy adopts 10 SDGs among the 17 SDGs from United Nations, later called as Suwon SDG (SSDG). The authors carry out a content analysis to value policies, public services, city administration and management, the knowledge economy, governance, engagement, collaboration, the built environment and city infrastructure, human capital and creativity, information and communication technologies (ICT), the natural environment and ecological sustainability, and data and information. They find that the indicators adopted from SDGs require further investigations for the properness of the core SSCI, although the study also covers the SDGs.

I want to congratulate the author(s) for the work. The content is relevant and the paper is well-written and rigorous. I have just some minor issues:

Abstract: While the abstract is well-written, the results are not very clear and it appears to be somewhat lengthy.

Results: The section might be lengthy for the reader. Since this section is crucial, and to make the reading experience more reader-friendly and engaging, I suggest that the authors consider condensing the content.

Overall, I perceive the manuscript as somewhat lengthy. If it is possible, I suggest condensing the text and emphasizing the core aspects. This would enhance the fluidity of the reading.

Author Response

(The authors gave the same response as above.)

Reviewer 4 Report

1. While some properties were given about the city, the social-economic, demographic, geopolitic and climite properties of the analysed city can be given, and then discussed on the sustainability aspects.

2. some information in the results section can be moved to introduction or litreture review, for example, studies on the Suwon city, and city characteristics. Because it not include any results about the present study.

3. the findings from study is not clear so please add a scheme or conclusion related the clear findings 

4. maybe, some tables presented as supplementary.

The english of the MS is well, only some sentences is too long, so it needs simplified. Besides there are some formating and punctuality errors.

Author Response

(The authors gave the same response as above.)
